# A scoping review of the unmet needs of patients diagnosed with idiopathic pulmonary fibrosis (IPF)

**Carita Bramhill**[1]* , **Donna Langan**[2‡], **Helen Mulryan**[2‡], **Jessica Eustace-Cook**[3‡], **Anne-Marie Russell**[4], **Anne-Marie Brady**[1]

**1** Trinity Centre for Practice & Innovation, School of Nursing and Midwifery, Trinity College Dublin, Dublin, Ireland, **2** Respiratory Department, Galway University Hospital, Galway, Ireland, **3** Trinity College library, Trinity College Dublin, Dublin, Ireland, **4** Institute of Clinical Sciences, College of Medical and Dental Sciences (MDS) University of Birmingham, Birmingham, United Kingdom

☉ These authors contributed equally to this work.
‡ These authors also contributed equally to this work.
* bramhilc@tcd.ie

**Data Availability Statement:** All relevant data are within the paper and its Supporting Information

## Abstract

### Aims

Patients diagnosed with idiopathic pulmonary fibrosis (IPF) have a high symptom burden and numerous needs that remain largely unaddressed despite advances in available treatment options. There is a need to comprehensively identify patients' needs and create opportunities to address them. This scoping review aimed to synthesise the available evidence and identify gaps in the literature regarding the unmet needs of patients diagnosed with IPF.

### Methods

The protocol for the review was registered with Open Science Framework (DOI 10.17605/OSF.IO/SY4KM). A systematic search was performed in March 2022, in CINAHL, MEDLINE, Embase, PsychInfo, Web of Science Core Collection and ASSIA Applied Social Science Index. A comprehensive review of grey literature was also completed. Inclusion criteria included patients diagnosed with IPF and date range 2011–2022. A range of review types were included. Data was extracted using a data extraction form. Data was analysed using descriptive and thematic analysis. A total of 884 citations were reviewed. Ethical approval was not required.

### Results

52 citations were selected for final inclusion. Five themes were identified: 1.) psychological impact of an IPF diagnosis. 2.) adequate information and education: at the right time and in the right way. 3.) high symptom burden support needs. 4.) referral to palliative care and advance care planning (ACP). 5.) health service provision-a systems approach.

### Conclusion

This review highlights the myriad of needs patients with IPF have and highlights the urgent need for a systems approach to care, underpinned by an appropriately resourced multi-

files. The data has also been filed on a preprint repository MEDRxiv.

**Funding:** CB was supported by the Irish Research Council; Grant number: GOIPG/2022/56 https://research.ie/funding-category/postgraduate/ The funder did not play a role in the preparation of this manuscript.

**Competing interests:** The authors have declared that no competing interests exist.

disciplinary team. The range of needs experienced by patients with IPF are broad and varied and require a holistic approach to care including targeted research, coupled with the continuing development of patient-focused services and establishment of clinical care programmes.

# 1 Introduction

## 1.1 Background

Interstitial lung disease (ILD) describes a range of heterogeneous lung conditions characterised by inflammation and fibrosis of the lung interstitium [1,2]. In the last decade there have been significant advances in our understanding of the pathophysiology of ILDs and the introduction of treatments that have significantly changed the landscape for many patients. [3] A large proportion of patients diagnosed with ILD have pulmonary fibrosis (PF)—most commonly idiopathic pulmonary fibrosis (IPF), representing around 17–37% of all ILDs. [4] IPF is a chronic progressive and irreversible disease which can profoundly and devastatingly impact the physical and psychological well-being of individuals [5,6].

People living with IPF may experience debilitating symptoms, which vary in severity and disease course. Symptoms include progressively worsening breathlessness, impaired lung function, cough and fatigue, [7,8] with many patients and their carers experiencing anxiety and/ or depression. [9,10] This high symptom burden, [1,11] coupled with social isolation for some, along with an inability to perform daily activities and the adjustment to a reduced life expectancy (median survival being 2–5 years), can impact quality of life (QoL) [12].

Affecting predominantly older adults [13] the incidence of IPF increases with age and with higher rates seen in males. [14,15] Internationally there has been a lack of standardisation in diagnostic coding, leading to an estimated reported prevalence of IPF ranging from 7 to 1650 per 100 000 persons. [16] Patients living with a diagnosis of IPF have high unmet care needs and require a multi-disciplinary team approach to care which should include supports such as multi-disciplinary team discussion at the time of diagnosis [1,17].

Incongruence persists between the needs of patients with IPF such as accurate and timely diagnosis, [18] referral for lung transplantation assessment, [17,19] access to pulmonary rehabilitation [9,18] and the actual delivery of healthcare services to adequately meet these needs. Individual needs of patients with IPF are important and so a person-centred approach encompassing the multiple components of the wider healthcare delivery system is needed. Addressing unmet needs for patients with IPF will contribute to improved quality of life. [18,20] Several studies including systematic reviews have previously investigated the care needs and experiences of patients with IPF. [7,8,18,20–24] However, many pre- COVID-19 studies have a broadly hospital-based focus with minimal recognition of the changing landscape of healthcare delivery, including community-based care.

Addressing unmet needs particularly for patients with IPF is deemed to be a critical issue and may facilitate the prioritisation of health services for this patient group and ultimately lead to improved quality of life. [18,20] Comprehensively understanding the unmet needs of patients with IPF can promote informed decision-making regarding patients' ongoing care and recognition of patient preferences. Despite advancements in our understanding of the pathogenesis of the disease and the ongoing delivery of antifibrotic treatment, deficits in our understanding of the needs and research priorities of patients with IPF prevail. Addressing the unmet needs of patients with IPF and designing services and patient-centred solutions around what patients want is essential to the future development of care for patients with IPF.

This review was guided by a central question, which was to map the available evidence related to the unmet needs of patients living with a diagnosis of IPF. The central research question was developed after several meetings with patient and public partners (PPI) comprising of patients diagnosed with IPF, their carers and healthcare professionals who collectively (a) described their research priorities and (b) identified the multi-dimensional component of their unmet needs. This was an iterative process and over the course of three meetings the research question took shape and led to the development of the scoping review protocol.

## 1.2 Aim

This scoping literature review aimed to examine the unmet needs of patients living with a diagnosis of IPF.

## 1.3 Objectives

1. To synthesise the unmet needs of patients living with a diagnosis of IPF.

2. Define barriers and facilitators to meeting patients' needs.

3. Provide an overview of relevant concepts and terminology.

## 1.4 Registration

The protocol for this study has been registered at Open Science Framework with its unique identifying number DOI 10.17605/OSF.IO/SY4KM.

## 2 Methods

### 2.1 Eligibility criteria

For the purpose of this review, patient needs have been defined from (a) PPI group input and (b) built on earlier work which mapped patients' care needs identified by the European IPF patient charter. New and emerging needs that were identified in the literature, and which to date have not been included in the European IPF patient charter, were mapped within this scoping review [20].

We utilised the 'PCC' framework, population (adult patients with IPF), concept (unmet needs), and context (all healthcare settings) to define the search strategy inclusion criteria. [25] The PCC framework used to inform the search strategy is presented in Table 1. The 'PCC' mnemonic (population, concept, and context) is recommended as a guide to construct clear objectives and eligibility criteria for scoping review [26].

The core concept in this review is 'unmet needs'. Studies describing the unmet needs specifically of patients diagnosed with IPF were included. Inclusion and exclusion criteria are presented in Tables 2 & 3.

### 2.2 Patient and public involvement

Patient and public involvement (PPI) is recommended from the earliest research stages through to dissemination of the study findings. [27] Patient and public involvement

**Table 1. PCC Framework for search strategy development.**

| Framework | Element | Key Terms |
| --- | --- | --- |
| PCC | Population | Adults > 18 years of age who have a diagnosis of idiopathic pulmonary fibrosis. |
| | Concept | Healthcare needs. |
| | Context | All care settings. |

**Table 2. Scoping review inclusion criteria.**

| Inclusion Criteria: | |
|---|---|
| | Sources must relate to patients with IPF. |
| | Sources must be published between 2011-to 2022. |
| | Adults >18 years of age. |
| | All geographic areas. |
| | All care settings. |
| | There are no language limits. Google translate will be used to translate non-English sources. |
| | We will include all review types, including systematic, scoping and literature reviews, which describe the unmet needs of our patient group. |

throughout the various stages of research is a valuable component of research activity and can contribute to improved quality and relevance of research. [28] Patient and public representatives (n = 5) from the Irish Lung Fibrosis Association PPI group were involved in reviewing the research protocol for this scoping review. The stakeholders comprise of patient representatives, family members of patients diagnosed with IPF and experts and researchers in the field of IPF. The scoping review protocol was presented to the group for feedback and discussion during a zoom call meeting. The scoping review findings were also presented to healthcare professionals in the field of IPF in person for further opportunities for discussion. The involvement of PPI in this scoping review has enriched the review and reflected the importance of including those impacted by IPF in the process.

## 2.3 Scoping review of the evidence

This review was conducted in accordance with the Joanna Briggs Institute framework for scoping reviews and included the following steps: (i) identifying the research question, (ii) developing a search strategy, (iii) study selection and (iv) data analysis and presentation. [29] The Preferred Reporting Items for Systematic Reviews and Meta-Analyses (PRISMA-ScR) checklist guides the reporting quality of this review (S2 Table) [30].

## 2.4 Sources & searching

The search strategy was developed with the assistance of a medical research librarian (JEC) and externally peer-reviewed by a second librarian as per the Peer Review of Electronic Search Strategies (PRESS) guidelines. [31] Six databases were systematically searched including, Medline, CINAHL, APA PsycINFO, (EBSCO platform), Embase (Elsevier), and Web of Science (Core Collection) and ASSIA (Applied Social Science Index) (Proquest), on 14th November 2022 and updated on 12th January 2023 to identify studies that met the review's inclusion criteria. Date limit criteria was applied at full text review (2011-present). A restriction to literature published pre-2011 was applied as this was around the time that antifibrotic treatment for patients with IPF became available in the United Kingdom and Europe.

**Table 3. Scoping review exclusion criteria.**

| Exclusion criteria: | |
|---|---|
| | Sources published prior to 2011. |
| | Refers to human subjects < 18 years of age. |
| | Does not include reference to patients diagnosed with IPF. |
| | This review will not include case reports, protocols, letters, commentaries or opinion pieces. |

No language or geographic limits were applied. Grey literature and unpublished studies were included; sources included ProQuest Dissertations and Thesis Global, Google Scholar and ClinicalTrials.gov, WHO International, Clinical Trials Registry Platform and OpenGrey. Further, a comprehensive online search of key websites and a manual search of the reference lists of included studies was performed. Several international conference abstracts were reviewed including those from the Irish thoracic society annual scientific meeting, the European respiratory society annual meeting, the British thoracic society meetings, and the American thoracic society meeting. A total of 100 abstracts were reviewed with 9 abstracts included in the final review.

The search strategy and database search were both conceptualised by the researcher (CB) and an information specialist (JEC). Key search terms related to 'IPF', 'unmet needs', 'idiopathic pulmonary fibrosis' and 'pulmonary fibrosis' (The search string is available in S3 Table).

An example of a search completed on CINAHL (EBSCO) of the search terms to identify the population is included here using the 'PCC' acronym and specifically looking at the 'population' component of PCC, (MH "Idiopathic Pulmonary Fibrosis") OR (MH "Idiopathic Interstitial Pneumonias+") OR (MH "Pulmonary Fibrosis+") OR TI ("Idiopathic pulmonary fibros*" OR "Idiopathic interstitial pneumonia*" OR "Familial Idiopathic Pulmonary Fibrosis*" OR "Usual Interstitial Pneumon*" OR "fibrosing interstitial lung disease" OR "progressive fibrosis" OR "nonspecific interstitial pneumonia" OR "pulmonary fibros*") OR AB ("Idiopathic pulmonary fibros*" OR "Idiopathic interstitial pneumonia*" OR "Familial Idiopathic Pulmonary Fibrosis*" OR "Usual Interstitial Pneumon*" OR "fibrosing interstitial lung disease" OR "progressive fibrosis" OR "nonspecific interstitial pneumonia" OR "pulmonary fibros*").

## 2.5 Study selection

Initially all identified records were collated and uploaded onto EndNote X9.3.3 (Clarivate Analytics, Pennsylvania, USA) and duplicates removed. Then all identified citations were transferred into Covidence software (Veritas Health Innovation, Melbourne, Australia) where any remaining duplicates were removed.

The next step was to screen the titles and abstracts conducted by two independent reviewers (CB and DL) for assessment against the inclusion criteria. Potentially relevant studies which met the inclusion criteria were retrieved in full text and uploaded to Covidence. The full text of selected citations was assessed in detail against the inclusion criteria (PCC inclusion criteria) by the same two independent reviewers. Any disagreements which occurred about the inclusion or exclusion of a paper were discussed by the reviewers until agreement was found. A third reviewer HM arbitrated when there was disagreement about the inclusion of a paper (n-4). We included all review types (systematic, scoping and narrative reviews) which described the unmet needs of our patient group. This review did not include case reports, protocols, letters, commentaries or opinion pieces.

## 2.6 Data extraction

Data was extracted from articles and other evidence sources included in the scoping review by one reviewer (CB), using a data extraction tool developed by the study's research team in adherence with the review objectives (S4 Table). All studies received verification by another reviewer (AMB & AMR). Several important domains were included in the data capture tool which adhered to the JBI guidelines.

## 2.7 Data analysis

Data were analysed utilizing the Braun and Clarke (2006) framework for reflexive thematic analysis (TA) [32]. This is a five-stage process for coding and data analysis and includes an

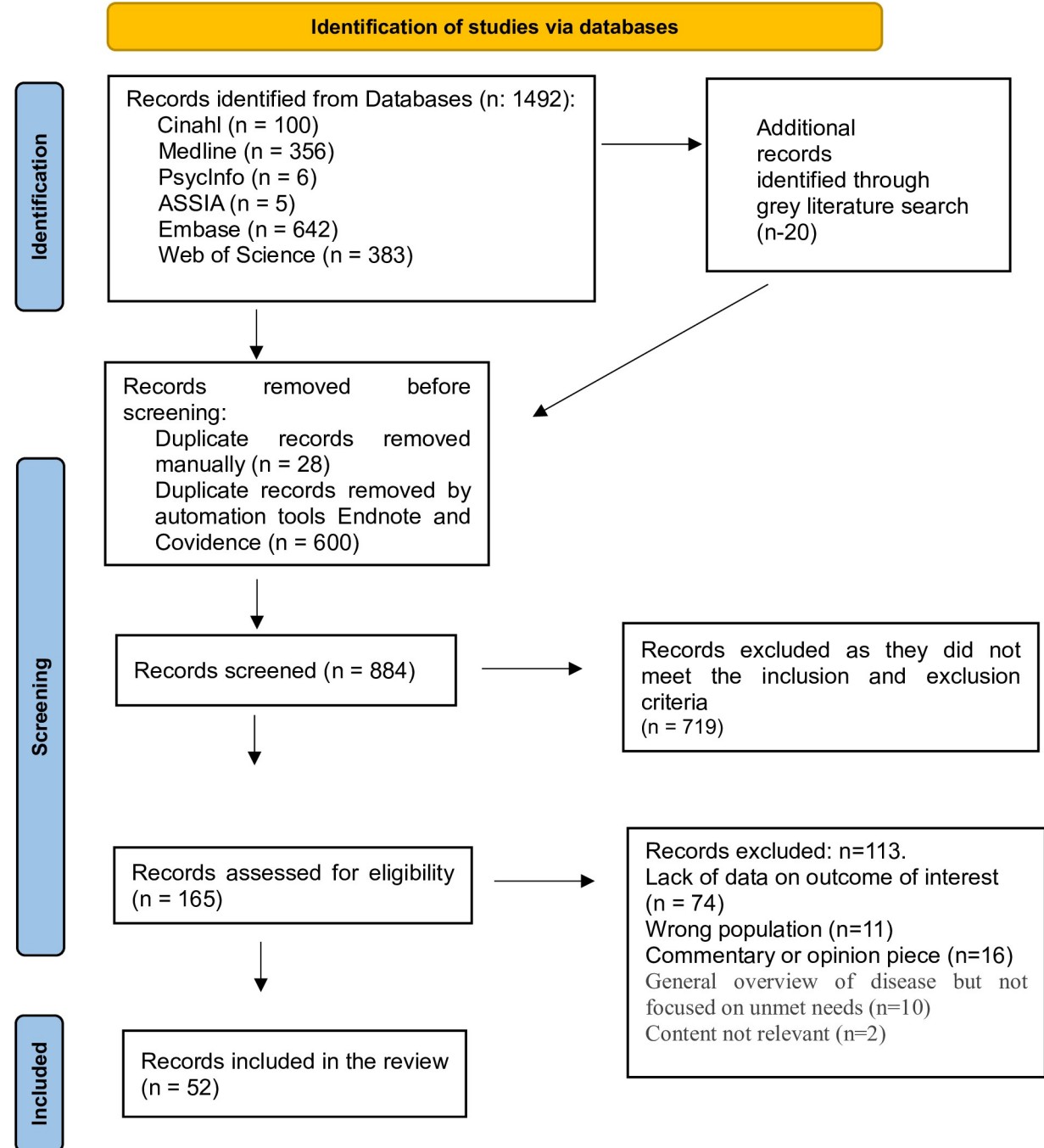

**Fig 1. Preferred reporting items for systematic reviews and meta-analyses (PRISMA).**

initial step of data familiarisation through deep immersion with the information sources. Next, an iterative process of code development was supported by analysis software NVivo (NVivo version 12.0). [32] The final three steps provided the backdrop to the organisation of codes into themes and fostered a rich in-depth analysis of the data, culminating in theme development. The preferred reporting items for systematic reviews and meta-analyses (PRISMA) displays the identified sources (Fig 1).

## 3. Results

### 3.1 Summary of the studies

A total of 1492 information sources were identified through our database search, and a further 20 records identified through a grey literature search, with 628 duplicates removed through manual and automated tools. After eliminating duplications, 884 unique citations were identified with five of these representing non-English sources. Of these, 719 records were excluded after title/abstract screening, leaving 165 records for further assessment. After full text review, 113 records were deemed ineligible and were excluded. The primary reason for exclusion was the lack of data on the outcome of interest (n = 74). Other common reasons for exclusion were wrong population (n = 11) and commentary or opinion piece (n = 16). Fifty-two information sources met the inclusion criteria, of which (n = 30/58%) were published in the last five years. All included articles were published between the period of 2011 to 2022. Included sources were published in English and were from a wide geographical area, including several countries where English is not the dominant spoken language and presented in Fig 2.

There was a total of thirty-eight studies included in this review, representing 73% of included records, with the remaining (n = 14/27%) records being a mix of literature reviews (n = 6), reports and guideline-type material, incorporating various methods (n = 8).

Of the studies (n = 38/73%), 50% of them employed quantitative approaches (n = 19), with others using a qualitative design, 37% (n = 14), while further studies employed a mixed method methodology 10.5% (n = 4). Several literature reviews (n = 6/11.5%) were included in the overall review of fifty-two records and incorporated a range of methods. The remaining information sources (n = 8/15.3%) were diverse and included guideline documents (n = 2), patient charters (n = 2), framework documents (n = 1) reports (n = 2) and a position statement (n = 1).

Of the thirty-eight studies (73%) included in the review, >31.6% (n = 12) of these investigated care experiences whilst a further 65.7% (n = 25) explored palliative care needs and advanced care planning. The remaining key areas of focus included in the thirty-eight studies predominantly explored information needs 71% (n = 27), pharmaceutical treatments 65.8%

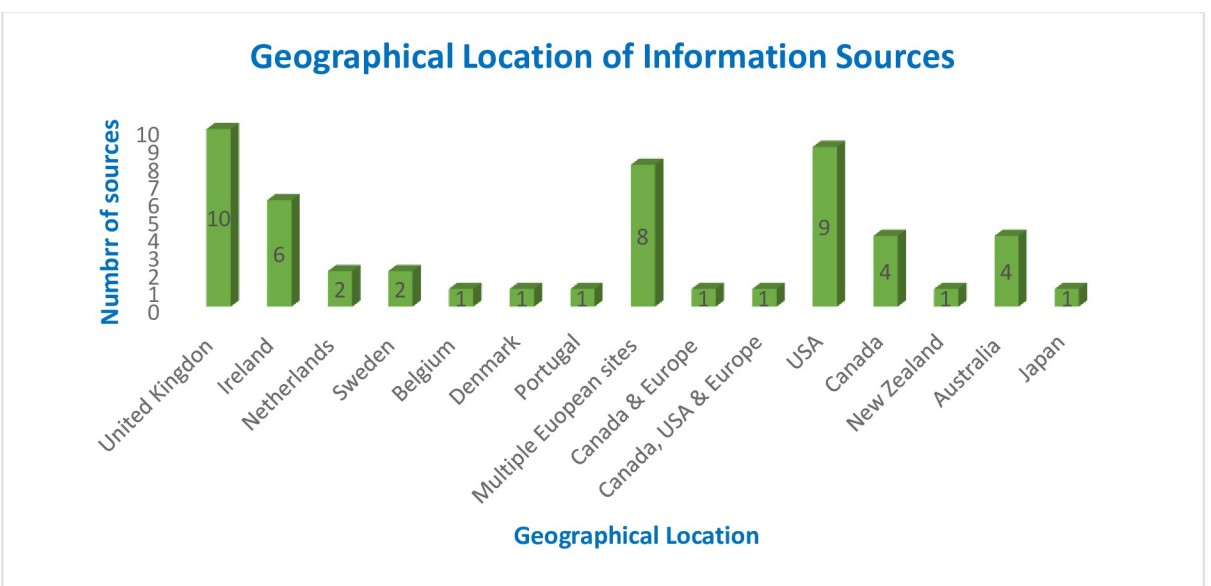

**Fig 2. Geographical location of information sources.**

(n = 25), early and accurate diagnosis 47.3% (n = 18) and psychological and emotional needs 52.6% (n = 20). Other significant areas included the impact on relationships 21.1% (n = 8), physical burden of IPF 36.8% (n = 14), oxygen needs 31.6% (n = 12), pulmonary rehabilitation 36.8% (n = 14), carer experience 39.5% (n = 15), geographical location 15.8% (n = 6), access to a tertiary centre 23.7% (n = 9), multi-disciplinary team 15.8% (n = 6), ILD nurse 23.7% (n = 9), tele-health 10.5% (n = 4), lung transplantation 31.6% (n = 12) and clinical trial access 7.9% (n = 3).

Recruitment strategies within the literature were reported to be via single sites 39.4% (n = 15) or multiple ones ranging from two to fourteen sites 60.5% (n = 23). A homogenous IPF sample was investigated in most sources (84%). Remaining studies featured heterogenous samples and included other ILD's, specifically fibrosing interstitial lung disease (F-ILD) (5.2%), pulmonary fibrosis (8.2%) and ILD (2.6%).

Male participants were significantly more frequently represented in all included studies (range 41.3%-87.7%). The fact that males were more frequently represented in our selected studies is aligned with international evidence that IPF is a disease predominantly seen in males (30) accounting for ~70% of all IPF cases in international cohorts, which may in part be explained by the fact that men are more likely to present with a smoking history and having experienced occupational exposures more frequently than women with IPF. (31) Study participants had an age range of 20–90 years. Studies which reported ethnicity or race of participants included white backgrounds (n = 5/13.1%). Only a limited number of studies included information on employment status (n = 2/ 5.2%), education (n = 4/10.5%), insurance (n = 3/7.8%) and marital status (n = 1/2.6%), a summary of these results is presented in Table 4.

## 3.2 Reflexive thematic analysis

There were several unmet needs and gaps in care identified in the literature and presented in Table 5.

**Table 4. Summary of the characteristics of the included articles.**

| Characteristics | N | % |
|---|---|---|
| **Methodology** | **(N-52)** | |
| Quantitative | 50 | 36.5% |
| Qualitative | 14 | 26.9% |
| Mixed methods | 4 | 7.7% |
| RCT | 1 | 1.9% |
| Review | 6 | 11.5% |
| Other | 8 | 15.3% |
| **Design** | **(N-38)** | |
| Cross sectional | 26 | 68.4% |
| Cohort | 5 | 13.1% |
| Longitudinal | 7 | 18.4% |
| **Methods of data collection** | **(N-38)** | |
| Questionnaire | 13 | 34.2% |
| Semi-structured interviews | 9 | 23.6% |
| Focus group | 4 | 10.5% |
| Registry data | 4 | 10.5% |
| World cafe | 1 | 2.6% |
| Chart Review | 2 | 5.2% |
| Mixed methods | 5 | 13% |

**Table 5. Unmet needs of patients diagnosed with IPF identified in the included evidence sources.**

| | | Unmet needs of patients diagnosed with IPF identified in the included evidence sources. |
|---|---|---|
| **Carer experience** | 13 | Kalluri et al., 2021, [33] Russell, Ripamonti and Vancheri, 2016, [34] Overgaard et al., 2016, [23] Cassidy et al.,2021, [35] Ramadurai et al., 2018a, [36] Meadows et al., 2017, [37] Masefield et al., 2019, [38] Giot, Kirchgassler and Maronati, 2012, [39] Sampson et al., 2015, [6] Tikellis et al., 2020, [21] Bajwah and Yorke, 2017, [15] Lee et al., 2020, [21] Lindell et al., 2021.[40] |
| **Pulmonary rehabilitation** | 15 | Moor et al., 2019, [18] Weatherald, McFadden and Fell, 2017,[5] Delameillieure et al., 2021,[41] Duck et al., 2015b, [42] Burnett, Glaspole and Holland, 2019,[43] Bonella et al., 2016, [20] Masefield et al., 2019, [38] EU-IPFF, 2020,[44] Sampson et al., 2015,[6] Tikellis et al., 2022, [17] Kalluri, Luppi and Ferrara, 2020, [12] NICE, 2017, [45] ITS, 2018, [46] NICE, 2015, [47] ILFA, 2015. [48] |
| **Information needs** | 24 | Moor et al., 2019, [18] Brereton et al., 2020, [49] Cove et al., 2015,[50] Ncube, 2020, [51] van der Sar et al., 2021,[52] Cassidy et al., 2021, [35] Masefield et al., 2019, [38] Overgaard et al., 2016, [23] Russell et al., 2016, [34] Meadows et al., 2017, [37] Ramadurai et al., 2018b, [13] Kalluri et al., 2022, [53] Delameillieure et al., 2021,[11] Burnett et al., 2019, [43] Bonella et al., 2016, [20] Sampson et al., 2015,[6] Tikellis et al., 2020, [54] Maher et al., 2018, [55] Kalluri et al., 2020, [12] Van Manen et al., 2017, [56] Bajwah and Yorke, 2017, [15] Lee et al., 2020, [21] ILFA, 2015, [48] Lindell et al., 2021.[40] |
| **Psychological/ emotional needs** | 19 | Cove et al., 2015, [50] Moor et al., 2019, [18] Giot et al., 2012, [39] Delameillieure et al., 2021, [41] Duck et al., 2015b, [42] Russell et al., 2016, [34] Overgaard et al., 2016, [23] Cassidy et al., 2021, [35] Schoenheit, Becattelli and Cohen, 2011, [57] Masefield et al., 2019, [38] van der Sar et al., 2021,[52] EU-IPFF, 2020, [44]Tikellis et al., 2022,[17] Kalluri et al., 2020,[12] van Manen et al., 2017,[56] Bajwah and Yorke, 2017,[15] Lee et al., 2020,[21] ILFA, 2015,[48] Lindell et al., 2021.[40] |
| **Physical burden of IPF** | 14 | Lancaster et al., 2022,[8] Turnpenny, Shepherd and Borrill, 2015,[58] Duck et al., 2015b,[42] Schoenheit et al., 2011,[57] Burnett et al., 2019,[43] Giot et al., 2012, [39] Sampson et al., 2015,[6] Kalluri et al., 2020, [12] van Manen et al., 2017, [56] Bajwah and Yorke, 2017, [15] Chaaban et al., 2022,[59] Lee et al., 2020, [21] ITS, 2018,[46] NICE, 2017.[45] |
| **Palliative care and advance care planning** | 23 | Akiyama et al., 2020, [60] Moor et al., 2019, [18] Sharp et al., 2018,[61] Weatherald et al., 2017,[5] Moor et al., 2021, [62] Ahmadi et al., 2016, [63] Maher et al., 2017, [64] Tyas, Boland and Gillon, 2019,[65] Turnpenny et al., 2015,[58] Kalluri et al., 2022,[53] Cassidy et al., 2021,[35] Delameillieure et al., 2021, [41] Overgaard et al., 2016,[23] Lindell et al., 2017, [7] Bonella et al., 2016, [20] Masefield et al., 2019, [38] EU-IPFF, 2020,[44] Kalluri et al., 2020, [12] van Manen et al., 2017,[56] Chaaban et al., 2022,[59] ILFA, 2015,[48] ITS, 2018,[46] Lindell et al., 2021.[40] |
| **Impact on relationships** | 8 | Lancaster et al., 2022, [8] Duck et al., 2015b,[22] Russell et al., 2016,[34] Overgaard et al., 2016b, [23]Lindell et al., 2017, [7] Schoenheit et al., 2011,[57] Sampson et al., 2015, [6]Lindell et al., 2021.[40] |
| **Oxygen needs** | 11 | Maher et al., 2017,[64] Duck et al., 2015b, [22]Overgaard et al., 2016, [23] Burnett et al., 2019, [43] Bonella et al., 2016, [20] Meadows et al., 2017,[37] EU-IPFF, 2020, [44] Tikellis et al., 2022, [17] van Manen et al., 2017, [56] Bajwah and Yorke, 2017, [15] ILFA, 2015.[48] |
| **Early & accurate diagnosis** | 19 | Lancaster et al., 2022, [8] Moor et al., 2019, [18]Weatherald et al., 2017, [5]Maher et al., 2017,[64] Brereton et al., 2020, [49] Schoenheit et al., 2011,[57] Delameillieure et al., 2021, [41] Overgaard et al., 2016,[23] Lindell et al., 2017, [7] Burnett et al., 2019, [43] Bonella et al., 2016,[20] van der Sar et al., 2021, [52] Duck et al., 2015b,[22] EU-IPFF, 2020,[44] Sampson et al., 2015,[6] Kalluri et al., 2020, [12] ITS, 2018,[46] ILFA, 2015, [48] NICE, 2017.[45] |
| **Care experiences** | 9 | Ncube, 2020,[51] Lancaster et al., 2022,[8] Moor et al., 2019,[18] Burnett et al., 2019,[43] Delameillieure et al., 2021,[41] Russell et al., 2016, [34] Giot et al., 2012, [39] Sampson et al., 2015,[6] Kalluri et al., 2020.[12] |

*(Continued)*

**Table 5.** (Continued)

|  |  | Unmet needs of patients diagnosed with IPF identified in the included evidence sources. |
|---|---|---|
| **Pharmaceutical treatment** | 22 | Lancaster et al., 2022, [8]Pesonen et al., 2018, [66]Picavet et al., 2017, [67] Maher et al., 2017,[64] Brereton et al., 2020,[49] Moor et al., 2019, [18] Weatherald et al., 2017,[5] Ncube, 2020,[51] Burnett et al., 2019,[43] Delameillieure et al., 2021,[41] Russell et al., 2016,[34] Bonella et al., 2016, [20] Masefield et al., 2019,[38] Maher et al., 2018,[68] Tikellis et al., 2022, [17] Lee et al., 2020, [21] van Manen et al., 2017,[56] Robalo-Cordeiro et al., 2017, [69] Bajwah and Yorke, 2017,[15] ILFA, 2015,[48] ITS, 2018,[46] NICE, 2017.[45] |
| **Geographical location** | 6 | Dedent, Collard and Thakur, 2021, [70] Johannson et al., 2022,[71] Swaminathan et al., 2022,[72] Ncube, 2020, [51] Cassidy et al., 2021,[35] Burnett et al., 2019.[43] |
| **Multi-disciplinary team** | 6 | Moor et al., 2019, [18] Maher et al., 2017, [64] Delameillieure et al., 2021,[41] Burnett et al., 2019, [43] Tikellis et al., 2022, [17]ITS, 2018.[46] |
| **E-health** | 3 | Ramadurai et al., 2018,[13]Tikellis et al., 2020,[54] Tikellis et al., 2022.[17] |
| **Lung transplant** | 11 | Swaminathan et al., 2022,[72]Moor et al., 2019, [18] Weatherald et al., 2017, [5] Cassidy et al., 2021,[35] Burnett et al., 2019, [43] Bonella et al., 2016, [20] Masefield et al., 2019, [38] Tikellis et al., 2022,[17] NICE, 2017,[45] ITS, 2018, [46] ILFA, 2015.[48] |
| **ILD Nurse** | 8 | Moor et al., 2019,[18] Delameillieure et al., 2021,[11] Russell et al., 2016,[34] Tikellis et al., 2022,[17] Kalluri et al., 2020,[12] NICE, 2017, [45] ITS, 2018, [46] Lindell et al., 2021.[40] |
| **Access to a tertiary care centre** | 8 | Lamas et al., 2011,[73] Maher et al., 2017, [64] Brereton et al., 2020,[49] Moor et al., 2019,[18] Dedent et al., 2021, [70]Weatherald et al., 2017,[5] Schoenheit et al., 2011,[57] Giot et al., 2012. [39] |
| **Clinical trial** | 3 | Maher et al., 2017, [64]Burnett et al., 2019, [43] Tikellis et al., 2020.[54] |

The process of reflexive thematic analysis enabled the development of five themes relating to the unmet needs of patients living with a diagnosis of IPF and presented in Fig 3.

## 3.3 The psychological and emotional impact of an IPF diagnosis

A need exists for psychological and emotional support throughout the disease course for patients diagnosed with IPF. [18,50] The literature reports that the psycho-social needs of patients [41] and their family carers are frequently being overlooked, [7,34] with a continued lack of psychological supports for patients diagnosed with IPF. [9,34] Psychological distress is reported by many patients living with IPF, including worry, fear, anxiety, hopelessness and helplessness. [7,12,23,34] It is reported that many patients with IPF and their carers experience anxiety and or depression [9,23,34,74].

Patients diagnosed with IPF report a loss of independence coupled with feelings of powerlessness and social isolation [23] The initiation of oxygen therapy is viewed as a particularly stressful time for patients [23], representing for some a distressing trajectory in the disease course and in some cases a loss of hope. [34] In one study oxygen initiation was associated with feelings of shame [41], as the condition became externally visible to others. [34] Patients can also experience increased anxiety related to worry associated with having adequate supplies of prescribed oxygen [41].

Glaspole and colleagues explored the frequency of prolonged anxiety and depression among people living with IPF and factors contributing to their persistence. They reported that dyspnoea is a major contributor to anxiety and depression followed by cough, which is also an important contributor. [75] Moor et al. found that although patients were not being specifically asked about access to psychological support in their study, 10% of patients spontaneously reported a need for improved psychological support throughout the disease course. [18] Van Manen and colleagues highlight the important role an ILD specialist nurse can play in helping

**Theme one**

☐ **Psychological/Emotional impact of an IPF diagnosis**

☐**Subthemes:** Psychosocial needs. Systemic impact on self and family members. The impact of anxiety, depression & stress. Emotional support, peer support, social isolation, coping strategies.

☐**Codes:** Psychological needs. Emotional well-being. Social needs.

**Theme two**

• **Adequate information and education, at the right time and in the right way**

☐**Subthemes:** Inadequate information provision. Consideration for timing of gradual and empathetic information about IPF. Patient advocacy groups/ role in information provision. Varied experiences of information and support particularly at time of diagnosis. Review understanding of disease trajectories regularly with the patient.

☐**Codes:** Information on living well.

**Theme three**

☐ **High symptom burden support needs**

•**Subthemes:** Influence of symptoms on well-being. Symptoms and co-morbidities increase complexity of care.  High symptom burden. Impact of medication side effects on patients. ☐ Recognition of supportive care needs. Impact of sexual dysfunction on relationships.

•**Codes:** Symptom experience . Functional needs . Physical needs.

**Theme four**

☐ **Referral to palliative care and advanced care planning**

•**Subthemes:** Earlier, open conversations with a specialist. Review goals and hopes. Prepare for end-of-life care with options. Preferred place of dying. Discuss options.

•**Codes:** Open and honest conversations about end-of-life care option.

**Theme five**

☐ **Health service provsion-an integrated systems approch**

☐**Subthemes:** Access to anti-fibrotic medication. Access to clinical trials. Regular ☐monitoring and care from a multi-disciplinary team including an ILD specialist nurse. ☐Burden of travel to access care. Appropriate access and referral to lung transplant assessment, specialist centres, national ILD registry, E-health, pulmonary rehabilitation.

☐**Codes:** Clinical care programme. Early and accurate diagnosis. Economic burden on healthcare system.

**Fig 3. Theme development.**

patients to manage symptoms such as depression and anxiety, particularly as nurses will most likely have been involved in the patients' care for some time and may be viewed by patients as someone, they can confide in [56].

## 3.4 Caregiver's support needs

Significantly the literature reported on the psychological impact of an IPF diagnosis on family caregivers who reported feelings of loneliness and anxiety [23,34], particularly associated with the fear of losing their loved-one. [23,74] Caregivers are often not adequately prepared to help their loved-one and describe a sense of frustration and helplessness. [74] Ramadurai and colleagues coined the concept "Shrinking world syndrome" in relation to caregivers to highlight the risk of social isolation, loneliness and a restricted lifestyle felt by some [13].

## 3.5 Adequate information and education at the right time and in the right way

Unmet information needs are prevalent for both patients with IPF and their carers in the presence of a varied disease trajectory. [6,20,41] The European Patient Charter calls for "comprehensive and high-quality information about IPF including its treatment to be made available to patients". *[20]*. Timely delivery of clinically appropriate information regarding diagnosis and treatment is an important cornerstone in the management of patients with a diagnosis of IPF. [46] For effective communication, patients and carers want plain language, honesty and empathy. [38] Furthermore, patients and carers want information with attention to timing, [38] content, [23] structure and format [6].

Practical information needs include information on medication use and potential side effects, [38] supplemental oxygen use, nutrition, exercise [42], management of cough and breathlessness [6], insurance cover, travel advice [particularly for those on oxygen], trusted online information resources [13], legal and practical advice for disease progression and end-of-life and palliative care planning for patients. [20,60] Patients living alone expressed the most direct and urgent need for information about future care planning. [6] There is an emerging need for information on research related to the outcomes of clinical trials for IPF. [13,24] Many patients and carers are not well informed about how their disease will progress. They require information on what to expect and how to prepare for the future [12,13,34,76] with an emphasis on an individualised approach [13,57].

It is understood that access to information and education from a diverse range of sources enhances patients' and carers' coping strategies. [35,56] Education and reliable information are the bedrock of patient care and help to empower patients to play an active role in their care [56].

Patients regularly turn to online sources of information on IPF, but these can be of poor quality, outdated or not available in the patients' native language. [34,77] Russell and colleagues found that the level of disease awareness varied extensively between patients and reported that approximately one third of patients felt inadequately aware of or informed about IPF [34].

Caregivers often felt inadequately prepared for their caregiving role and expressed a need for information and education on strategies to help their family member manage IPF; for some there was also a requirement for information on palliative care and advanced care planning (ACP) [34, 35].

Furthermore, there are increased calls for more awareness of IPF among GPs, nurses and physicians [18] and the general public. [20] Healthcare professionals have expressed concern over a lack of time to discuss the diagnosis and treatment options (60%), with only 39% of healthcare professionals reporting that they had received any training in patient-centred communication [18].

## 3.6 High symptom burden

A significant unmet need related to IPF is the burden associated with the physical and psychological impact of the condition. IPF remains an unpredictable disease of variable course which could benefit from a systems approach to care, coordinated by a multi-disciplinary team. [6] The deterioration in health-related quality of life for patients with IPF is highly correlated to worsening of symptoms, including increased breathlessness, cough and fatigue over time. [12] Lindell et al. analysed focus group data and highlighted that symptoms introduce an overwhelming burden for both the patient and carer, with cough being a particularly challenging symptom. [7] For some patients coughing led to distressing symptoms such as incontinence. [66] Patients report struggling with lethargy which can impact even the simplest of tasks, such as reading or watching television [22].

Several patients in Duck's study reported feeling depressed which was associated with a lack of control and having to relinquish roles once held. [22] There is growing evidence that daily activities, recreation, pleasure and employment are significantly affected by the burden of symptoms like anxiety, depression and social isolation that are connected with a diagnosis of IPF. [12] There were also reported symptoms from the side effects of medication—in particular antifibrotic treatment adding to the burden of symptoms already experienced by patients [43].

Sampson and colleagues call for a more pragmatic needs assessment to include components of physical and social functioning, nutrition and symptom burden which would support patients' self-management and assist with their understanding of the illness and its varied disease trajectories [6].

## 3.7 Health service provision—a systems approach

Reliance on healthcare services is immense for those living with IPF, with high healthcare costs in terms of resourcing and utilisation of services, provision of multi-disciplinary care and a recognised marked socio-economic burden for patients. [8,78] Patients with IPF require regular routine monitoring and input from the multi-disciplinary team, including provision for repeated hospitalisation. There is a need for increased supportive care particularly at the end-of-life [8].

A major unmet need in IPF care is the provision of timely and accurate diagnosis [13,18,22] and this is a recurrent deficit in health service provision. Several studies have highlighted the scope of the problem, with many patients diagnosed with IPF experiencing a protracted route to diagnosis [7,18,20,22,41,43,44, 52] which leads to unnecessary delays including accessing pharmacological interventions and other supportive treatments [79].

There is an increasing recognition of the importance of having a well-resourced and appropriately staffed multi-disciplinary team (MDT) in providing care to patients with IPF. [17] Current guidelines call for multi-disciplinary discussion involving expert respiratory physicians, radiologists and pathologists as the gold standard for IPF diagnosis. [1] However, evidence exists which suggests that there can be gaps in staffing some of these multi-disciplinary teams [17,80].

Patients should have equal opportunities to access a variety of healthcare professionals encompassing a holistic approach to care and optimising their quality of life, with calls for equal access to non-pharmacological treatments options including pulmonary rehabilitation, psychological support and transplant assessment referral. [20] Many studies continue to highlight that the myriad of non-pharmacological treatment options are not equally available for patients in different geographical locations [20,81].

Clinical nurse specialists are an essential component of the multi-disciplinary team and are critical to the delivery of holistic care. [22,46,47] Clinical nurse specialists play an essential role

as part of a multi-disciplinary team through several key channels of care, including supporting patients with IPF through the provision of expert knowledge and advice throughout the disease course [22,34] and coordinating care within the multi-disciplinary team. [82] However, there can be variability in access to specialist nursing across jurisdictions. [81,83] Furthermore, access to other supportive care services such as pulmonary rehabilitation [12] can also be fragmented with one study reporting that just 42% of patients had access to outpatient pulmonary rehabilitation, with similar findings reported for access to psychological support (58%) [18].

From a systems approach to healthcare there is an urgent need for the establishment and recognition of national registries to capture epidemiological information on patients diagnosed with IPF. [46,66] There is also a need reported by healthcare professionals to recognise the importance of giving patients the option of participating in research or clinical trials related to IPF [18,22].

### 3.8 Referral to palliative care and ACP

The World Health Organization recommends early palliative care intervention to improve the quality of life of patients and their families facing problems associated with a life-threatening illness. Since the course of IPF is unpredictable, early palliative care interventions can be beneficial in a multiplicity of ways including symptom management, [20,84] emotional support and in the initiation of advance care planning conversations [59].

The European IPF Patient Charter and the Irish Thoracic Society Position Statement on the management of IPF have identified an urgent need to involve palliative care in IPF. [20,35] The National Institute of Health and Care Excellence (NICE) guidance is that patients with IPF should have access to palliative and supportive care services to manage symptoms. [47,61] Startlingly, despite these recommendations, patients with IPF do not receive optimal palliative care over the course of their disease, resulting in high symptom burden and decreased quality of life for patients. [59,84] There is increasing evidence that patients with IPF do not always have access to palliative care input and when it is introduced the timeline is rarely optimal. [18,35] There continues to be poor referral and access to palliative care specialists, with some healthcare professionals reporting a lack of training in palliative care, [59] insufficient communication training in facilitating end-of-life conversations, variations in the disease trajectory [60] and patients' preferences to have these conversations. [7,53,85] In one study healthcare professionals—specifically practising pulmonologists and nurses with ILD expertise—reported that palliative care was not initiated until the later stages of pulmonary fibrosis/IPF, with a fifth reporting that palliative care was only initiated at end-of-life. [18] Lack of advance care planning leads to longer Intensive care unit and overall hospital lengths of stay [86].

Kalluri and colleagues in their qualitative study revealed that advance care planning is desired by patients and caregivers early in their illness experience, with healthcare professionals citing a need to clarify role, scope and responsibility with a call for practical guidance and training for healthcare professionals to improve competency and confidence in these conversations [53].

## 4. Discussion

This scoping review characterised the broad and varied unmet needs of patients living with a diagnosis of IPF. These unmet needs spanned five core domains incorporating physical and psychological needs, palliative care needs and finally needs related to information and health service provision (S5 Table).

In recent years our knowledge of the pathogenesis of IPF, coupled with an improved awareness of the complex disease burden associated with this disease, has rapidly improved;

nevertheless, there remains a myriad of unmet needs and gaps in care for this patient population, which is reflected in several previous studies on the topic [6,18,20].

Despite advancements in drug therapies including the use of antifibrotic medication, which when first introduced heralded improved survival rates for many, there continues to be unmet needs and compromised quality of life for many patients living with IPF. [21] A key element of the inequalities and gaps in care that continue to exist for many patients is timely access to these antifibrotic medications [87] with continued barriers to treatment such as reimbursement restrictions evidenced internationally, despite the various calls which support early initiation of these treatments [18].

### 4.1 The psychological and emotional impact of an IPF diagnosis

Psychological distress was reported by many patients living with IPF and their caregivers and incorporated several key elements including anxiety, depression, fear and worry. [7, 23, 34] The continued need for improved access and reimbursement to psychological and emotional support throughout the disease course for both patients and their caregivers was evident throughout the literature. [22,23,34,35,38,39,41,44,52,57] The psycho-social needs of patients and their family carers represent a critical issue as regards the incorporation of psychological support into existing and newly developed clinical care pathways for patients with IPF, with a focus on eliminating cost and access issues to this important resource. The literature supports a broad scope of adjunct psychological supports for patients including the requirement for a respiratory nurse specialist, peer support programmes and pulmonary rehabilitation. [54] However, there remains continued disparate access to these resources internationally [41,54].

### 4.2 Adequate information and education at the right time and in the right way

Patients have specific requirements regarding the delivery of appropriate and bespoke information at specific time points during the disease trajectory, a need which remains largely unmet. [6,20,57] Support groups and patient organisations were highlighted as a potential resource for information delivery in conjunction with the patient's hospital-based team during outpatient clinic visits. Addressing information needs has been identified as a key cornerstone in international patient charters, but a gap between information needs and their actual delivery remains [20,48].

### 4.3 High symptom burden

Patients living with IPF report a high symptom burden impacting both physical and psychological well-being, coupled with management of drug side-effects, and a call for support in managing these disabling symptoms. The literature reported that there are further unmet needs for patients with IPF, not least the need for appropriately resourced multi-disciplinary teams [17]—including the requirement for respiratory nurse specialists [22,46].

Furthermore, access to specialist interstitial lung disease centres with bespoke expertise to help manage patients' symptoms and facilitate access to supportive care such as pulmonary rehabilitation is a requirement for patients with IPF.

### 4.4 Referral to palliative care and ACP

A critical unmet need for patients with IPF is timely referral to palliative care as part of their overall management plan and care package. Despite calls from several patient charters and international IPF position statements [20,35] highlighting the importance of access to

palliative and supportive care services to manage symptoms, [47,61] there continues to be poor palliative care referral and access to palliative care specialists, directly impacting patients' quality of life [59,84].

## 4.5 Health service provision -a systems approach

Patients living with IPF rely heavily on the healthcare system, representing a significant need for an integrated health systems' approach to providing supportive care. Equal access to supportive care and non-pharmacological treatment remains problematic, with reported unequal availability of services for patients in different European countries. [20,81] Internationally and across several healthcare systems there remain deficits in providing patients with an early and accurate diagnosis [13,18,22], representing a recurrent gap in health service provision. These delays have significant impacts to patients, including accessing pharmacological interventions and other supportive treatments [79].

In summary, the findings of this review extend our current understanding of the broad scope of unmet needs for patients with IPF, expanding our depth of awareness of patient experiences and requirements for support and access to essential services across multiple domains. Equally it identifies the diverse and nuanced approach to care which is required for this patient population. Furthermore, it identifies an urgent unmet need globally which is the call for the establishment of IPF patient registries to capture epidemiological information, [46,66], coupled with additional benefits such as ease of identification of patients who may wish to participate in research or clinical trials related to IPF. [18,22] Registry data will offer opportunities to improve our understanding of this complex condition and to adequately prepare for appropriately resourced care provision.

## 4.5 Study limitations

The core limitations of this scoping review involve the diverse and varying quality of available evidence with only one randomized controlled trial included. Similarly mixed methods studies were underrepresented in the review, comprising just 7.6% of the overall included resources. Furthermore, there were limited longitudinal studies, with just 18.4% represented in this review. The methodological quality of studies was high, yet many excluded some important demographic variables including stage of disease, oxygen use, socioeconomic status and employment status. Given the varied disease trajectory, the interpretation of the findings from some studies may have been impacted. This review included non-English language articles in its inclusion criteria but cannot claim to have exhausted all non-English resources despite utilising a systematic approach to the review.

## 4.6 Conclusion

In recent years we have seen significant advances in relation to our understanding of the pathogenesis of IPF, coupled with the introduction of antifibrotic medication and their recognised contribution to patient survival. The concept of unmet needs and quality of life are intrinsically linked and yet there remains deficits in the literature as regards comprehensively investigating this relationship for patients with IPF, representing a need for future research focus examining this relationship.

This review will extend the knowledge base of the muti-disciplinary team as regards the diverse range of needs that patient with IPF have and signals the need to continue to target research toward this underrepresented patient population. The literature highlights the continued lack of integrated clinical care programmes in many jurisdictions for the management of IPF, which can result in unstructured and fragmented care delivery for patients. This study

also highlights that patients living with a diagnosis of IPF experience a diverse scope of unmet needs across a broad range of areas and require a comprehensive multi-disciplinary approach to care, with equal access to services and tailored information to support them over the course of the disease. These are key areas for future research.

## Supporting information

**S1 Table. Glossary.**
(PDF)

**S2 Table. Prisma-ScR checklist.**
(PDF)

**S3 Table. Search of CINAHL (EBSCO) conducted on 14th November 2022.**
(PDF)

**S4 Table. Data extraction instrument.**
(PDF)

**S5 Table. Summary of the characteristics of included information sources.**
(PDF)

## Acknowledgments

With special thanks to the participants of the Irish Lung Fibrosis Association PPI group.

## Author Contributions

**Conceptualization:** Carita Bramhill, Jessica Eustace-Cook, Anne-Marie Russell, Anne-Marie Brady.

**Formal analysis:** Anne-Marie Russell.

**Methodology:** Donna Langan, Helen Mulryan, Anne-Marie Russell, Anne-Marie Brady.

**Resources:** Anne-Marie Brady.

**Supervision:** Anne-Marie Brady.

**Writing – original draft:** Carita Bramhill.

**Writing – review & editing:** Anne-Marie Brady.

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
