## [Decision Letter · Decision Letter 0]

10 Oct 2023

PONE-D-23-27557A scoping review of the unmet needs of patients diagnosed with idiopathic pulmonary fibrosis (IPF).PLOS ONE

Dear Dr. Bramhill,

Thank you for submitting your manuscript to PLOS ONE. After careful consideration, we feel that it has merit but does not fully meet PLOS ONE’s publication criteria as it currently stands. Therefore, we invite you to submit a revised version of the manuscript that addresses the points raised during the review process.

ACADEMIC EDITOR: Please ensure that your decision is justified on PLOS ONE’s publication criteria and not, for example, on novelty or perceived impact.

We look forward to receiving your revised manuscript.

Kind regards,

Zyad James Carr, MD

Academic Editor

PLOS ONE

Journal Requirements:

3. We note that you have referenced which has currently not yet been accepted for publication. Please remove this from your References and amend this to state in the body of your manuscript: (ie “Bewick et al. [Unpublished]”) as detailed online in our guide for authors

4. Please ensure that you include a title page within your main document. You should list all authors and all affiliations as per our author instructions and clearly indicate the corresponding author.

5. Please amend your manuscript to include your abstract after the title page.

Additional Editor Comments:

Hello Drs. Bramhill and Colleagues,

We want to thank you very much for the hard work of your narrative/scoping review regarding unmet needs of patients diagnosed with idiopathic pulmonary fibrosis. Below please find the recommended thoughtful revisions from the reviewers.

In addition, I would like to direct your attention to the following recommendations:

It would be of interest to the readership to briefly further expand on the diaspora of idiopathic pulmonary fibrosis as this diagnosis encompasses a vast number of subtypes.

You also discuss the incongruence between the needs of patients and actual delivery of healthcare services. An example would be fitting here.

Similarly, the introduction discusses informed decision making without appropriately framing the type of decision making. Please clarify.

Please further expand on the concept of the PCC framework and its evidence. Briefly.

Please apply 'restricted/trademark' symbology when appropriate.

Page 6 - please describe the exact percentage plus 38/xx of included studies. Please apply this concept through the manuscript.

Page 7 - you describe that male participants were significantly more frequently represented, it is important to include why - males often have more severe disease and more mortality, this may be a generalized selection bias or you may have other thoughts, please support this comment as you see fit.

The structure of the tables could be enhanced, both in aesthetic structure and brevity, consider revising to improve clarity, perhaps by including citations by last name and year of publication. Canada is incorrectly spelled in one instance. Abbreviations are applied w/out a glossary attached to the table.

Please expand Table 3 figure legend. Similarly, improve the aesthetic structure and clarity.

Page 13 - revise (better) - please either quote or appropriately paraphrase the study findings.

Page 17 - Please expand on the several studies highlight that nonpharmacological treatment options are...

Page 17 - please define clinical nurse specialists role as this may be foreign to the readership.

Page 17 - unclear why There is italicized?

Page 18 - World health organ... should be World Health Organization

Was HCP defined?

Ensure that all acronyms have been appropriately defined, e.g. ICU, or if there is adequate space/word count, just spell them out.

Page 19 Please define PPI group

Discussion

Please highlight the key findings in one sentence at the beginning of the discussion. Further details can be provided later. Given the complexity of the review, a point by point discussion is warranted, but first framed by a succinct topic sentence.

Psychological distress was heightened...why does this represent a critical issue in your words.

The findings of the review extend our current understanding...please explain further as to the specific met demands of the scoping review, e.g. characterizing X, identifying X, etc.

Reviewers' comments:

Reviewer's Responses to Questions

**Comments to the Author**

1. Is the manuscript technically sound, and do the data support the conclusions?

Reviewer #1: Yes

Reviewer #2: Partly

Reviewer #3: Yes

2. Has the statistical analysis been performed appropriately and rigorously? 

Reviewer #1: Yes

Reviewer #2: N/A

Reviewer #3: N/A

3. Have the authors made all data underlying the findings in their manuscript fully available?

Reviewer #1: Yes

Reviewer #2: Yes

Reviewer #3: Yes

4. Is the manuscript presented in an intelligible fashion and written in standard English?

Reviewer #1: Yes

Reviewer #2: Yes

Reviewer #3: Yes

5. Review Comments to the Author

Reviewer #1: Dear editor,

Thank you for providing me the opportunity to review this article titled “A scoping review of the unmet needs of patients diagnosed with idiopathic pulmonary fibrosis (IPF).” In this scoping review, authors have reported unmet needs of the patients with IPF. Authors have done a systematic review of literature and summarized the unmet needs into 5 themes. Overall, the manuscript is interesting and seems relevant. However, there are certain points in the manuscript which need clarification. I feel that clarification of these points will improve the quality of manuscript.

Please find below the points that need to addressed:

Eligibility criteria:

“The search was limited to human participants >18 years of age at the time of data collection (PCC Framework for search strategy development is available in appendix 1).”

IPF is disease of elderly, why the age >18 years set? Will it be better to replace it “adult”? Or studies that included young patient with pulmonary fibrosis were also included? This is important as the study title mention “IPF”.

2.2 Scoping Review of the evidence

“This review was conducted in accordance with the Joanna Briggs Institute framework for scoping reviews and included the following steps: (i)identifying the research question, (ii)developing a search strategy, (iii)study selection and (iv)data analysis and presentation. (22)”

The most important aspect of the systematic review is the identification/framing of research question. Therefore, authors should provide the information related to research question. How the research questions identified/framed? Who were involved in designing in research questions? Were there any modifications in research questions? If yes, how many revisions?

2.3 Sources & searching

“Further, a comprehensive online search of key websites, a manual search of the reference lists of included studies and a search of annual conference abstracts sought to identify studies that may have been missed within the initial search.”

Were all annual conferences searched? Or conferences were specific to Pulmonary Diseases, ILD, IPF? Also, were these only regional, national or international conferences?

How many abstracts were screened/included/excluded?

3.1 Summary of the studies

“All included studies were published in English and were from a wide geographical area (figure 2).”

There was no language barrier. However, authors did not report number of publications in language other than English.

Page 6; para 2: “Several literature reviews (n=6) incorporated a range of methods. The remaining information sources were diverse (n=8) including guideline documents (n=2), patient charters (2), framework documents (n=1) reports (n=2) and a position statement (n=1). Information sources originated from multiple sources.” The text seems misplaced.

“Kalluri and colleagues in their qualitative study revealed that advance care planning is desired by patients and caregivers early in their illness experience with HCPs citing a need to clarify role, scope and responsibility with a call for practical guidance and training for HCPS to improve competency and confidence in these conversations.” Please provide the reference of the study.

Conclusions

“This review included non-English language articles but cannot claim to have exhausted all non-English resources despite utilising a systematic approach to the review.”

However, there is no mention of the number of non-English studies in the manuscript. Also, if non-English studies were not excluded then how data extracted from those studies? Did authors take help from foreign language expert/s? If yes, please mention in the manuscript.

Reviewer #2: This scoping review by Carita Bramhill et al. explored the literature on idiopathic pulmonary fibrosis.

It examined 1492 studies, a large population that included only 52 of those studies.

My main concerns with this review are:

1- The inclusion of other reviews, it would be of more benefit to include studies from those reviews that are relevant to this current review.

2- The exclusion criteria are not specified so it is unclear how 1442 were filtered out.

3- The search terms in appendix 2 would help to understand point 2 if they were included in the manuscript.

Reviewer #3: The authors perform a scoping review to identify the gaps in clinical knowledge and recommendations related to the management of ILD. This is a critical area of focus to help to comprehensively improve the care of patients with ILD and is available addition to the literature. Some modifications to the manuscript would be of benefit to the value of the publication:

1) Can the authors provide a clear outline, perhaps in tabular format, of the inclusion and exclusion criteria for publications to be included in the study, as this remains unclear.

2) The study cohort should be better defined. The manuscript refers to patients with IPF and PF in various sections, and in the results section also includes ILD (presumably non fibrotic)? The population being studied therefore is unclear and makes interpretation difficult.

3) Can the authors clarify why only English language publications were ultimately included in the study?

4) The description in the results section related to patient and carer participation in study oversight seems better suited for inclusion in the methods.

5) Finally, the manuscript should be reviewed for grammatical errors (particularly the use of commas), which seem to be present throughout and detract from an otherwise very strong and well presented manuscript.

6. PLOS authors have the option to publish the peer review history of their article (what does this mean?). If published, this will include your full peer review and any attached files.

Reviewer #1: No

Reviewer #2: No

Reviewer #3: **Yes: **Hilary J. Goldberg, MD, MPH

---

## [Author Response · Author response to Decision Letter 0]

20 Nov 2023

This response has been uploaded as a table on PLOS one in my reply to the reviewer comments.

 Thank you for flagging this – the Plos One journal writing style has been applied throughout the manuscript.

 ttHNNThe pre-print for this scoping review is available at: https://www.medrxiv.org/content/10.1101/2023.09.11.23294619v1

We note that you have referenced which has currently not yet been accepted for publication. Please remove this from your References and amend this to state in the body of your manuscript: (ie “Bewick et al. [Unpublished]”) as detailed online in our guide for authors Thank you for this comment. We do not use a citation for Bewick in this paper nor reference any unpublished works. We do however cite conference abstracts where the abstract is published in a peer reviewed journal and where the full papers have not as yet been published e.g. refs 49 58 79 –please advise if you would like us to manage these differently.

 Please ensure that you include a title page within your main document. You should list all authors and all affiliations as per our author instructions and clearly indicate the corresponding author. Thank you - The title page is now included with all authors listed the corresponding author is clearly stated.

Please amend your manuscript to include your abstract after the title page Thank you – we have done this, and the abstract appears on the page after the title page.

Please include captions for your Supporting Information files at the end of your manuscript, and update any in-text citations to match accordingly. Please see our Supporting Information guidelines for more information: http://journals.plos.org/plosone/s/supporting-information. 

Thank you – we have now included captions for the supporting information files, which is listed on page 33 of the main manuscript.

In text citations are now updated.

It would be of interest to the readership to briefly further expand on the diaspora of idiopathic pulmonary fibrosis as this diagnosis encompasses a vast number of subtypes. Thank you – we agree that ILD is an umbrella term for approx. 200 disease entities. IPF is one of the most prevalent entities. Our background section now includes the following text:

‘Interstitial lung disease (ILD) describes a range of heterogeneous lung conditions characterised by inflammation and fibrosis of the lung interstitium. (1, 2) In the last decade there have been significant advances in our understanding of the pathophysiology of ILDs and the introduction of treatments that have significantly changed the landscape for many patients. (3) A large proportion of patients diagnosed with ILD have pulmonary fibrosis (PF) - most commonly idiopathic pulmonary fibrosis (IPF), representing around 17-37% of all ILDs. (4) IPF is a chronic progressive and irreversible disease which can profoundly and devastatingly impact the physical and psychological well-being of individuals. (5, 6)’

You also discuss the incongruence between the needs of patients and actual delivery of healthcare services. An example would be fitting here Thank you - our introduction section has been amended and now includes the following wording: 

‘Incongruence persists between the needs of patients with IPF such as accurate and timely diagnosis, (18) referral for lung transplantation assessment, (17, 19) access to pulmonary rehabilitation (9, 18) and the actual delivery of healthcare services to adequately meet these needs.’ 

Similarly, the introduction discusses informed decision making without appropriately framing the type of decision making. Please clarify Our Introduction section has been amended and now includes the following wording: 

‘Addressing unmet needs particularly for patients with IPF is deemed to be a critical issue and may facilitate the prioritisation of health services for this patient group and ultimately lead to improved quality of life. (18, 20) Comprehensively understanding the unmet needs of patients with IPF can promote informed decision-making regarding patients’ ongoing care and recognition of patient preferences’.

Please further expand on the concept of the PCC framework and its evidence. Briefly. We utilised the ‘PCC’ framework, population (adult patients with IPF), concept (unmet needs), and context (all healthcare settings) to define the search strategy inclusion criteria. (25) The PCC framework used to inform the search strategy is presented in Table 2. The ‘PCC’ mnemonic (population, concept, and context) is recommended as a guide to construct clear objectives and eligibility criteria for scoping review. (26) The core concept in this review is ‘unmet needs’. Studies describing the unmet needs specifically of patients diagnosed with IPF were included. Inclusion and exclusion criteria are presented in Table 3 & 4’. 

Please apply 'restricted/trademark' symbology when appropriate We were unable to detect where the ‘restricted/trademark’ symbology is overlooked or not applied.

Page 6 - please describe the exact percentage plus 38/xx of included studies. Please apply this concept through the manuscript. Thank you we have updated the manuscript to reflect your comments and now include both numbers and percentages for reported sources. 

Page 7 - you describe that male participants were significantly more frequently represented, it is important to include why - males often have more severe disease and more mortality, this may be a generalized selection bias or you may have other thoughts, please support this comment as you see fit Updated to include the following text: 

‘Male participants were significantly more frequently represented in all included studies (range 41.3%-87.7%). The fact that males were more frequently represented in our selected studies is aligned with international evidence that IPF is a disease predominantly seen in males (30) accounting for ~70% of all IPF cases in international cohorts, which may in part be explained by the fact that men are more likely to present with a smoking history and having experienced occupational exposures more frequently than women with IPF. (31)’

The structure of the tables could be enhanced, both in aesthetic structure and brevity, consider revising to improve clarity, perhaps by including citations by last name and year of publication. Canada is incorrectly spelled in one instance. Abbreviations are applied w/out a glossary attached to the table Thank you we have updated the tables to reflect your comments.

Please expand Table 3 figure legend. Similarly, improve the aesthetic structure and clarity Thank you we have updated the tables to reflect your comments.

Page 13 - revise (better) - please either quote or appropriately paraphrase the study findings Updated to include the following text:

‘Moor et al. found that although patients were not being specifically asked about access to psychological support in their study, 10% of patients spontaneously reported a need for improved psychological support throughout the disease course. (18)’ 

Page 17 - Please expand on the several studies highlight that nonpharmacological treatment options are.. Updated to include the following text: 

‘Patients should have equal opportunities to access a variety of healthcare professionals encompassing a holistic approach to care and optimising their quality of life, with calls for equal access to non-pharmacological treatments options including pulmonary rehabilitation, psychological support and transplant assessment referral. (20) Many studies continue to highlight that the myriad of non-pharmacological treatment options are not equally available for patients in different geographical locations. (20, 80)’

Page 17 - please define clinical nurse specialists role as this may be foreign to the readership Updated to include the following text: ‘Clinical nurse specialists are an essential component of the multi-disciplinary team and are critical to the delivery of holistic care. (22, 44, 45) Clinical nurse specialists play a critical role as part of a multi-disciplinary team through several key channels of care, including supporting patients with IPF through the provision of expert knowledge and advice throughout the disease course (22, 33) and coordinating care within the multi-disciplinary team. (81)’ 

Page 17 - unclear why There is italicized? Thank you, the text has been, updated and italics removed.

Page 18 - World health organ... should be World Health Organization Updated to include the following text:

World Health Organization

Was HCP defined? Updated to include the following text:

HCP removed and instead healthcare professional inserted and defined in the context of the aforementioned study.

‘In one study healthcare professionals - specifically practising pulmonologists and nurses with ILD expertise - reported that palliative care was not initiated until the later stages of pulmonary fibrosis/IPF, with a fifth reporting that palliative care was only initiated at end of life. (18) Lack of advance care planning leads to longer Intensive care unit and overall hospital lengths of stay. (85)’

Ensure that all acronyms have been appropriately defined, e.g. ICU, or if there is adequate space/word count, just spell them out. Updated to include the following text:

Acronyms removed and instead full text version of term inserted.

Page 19 Please define PPI group Updated to include the following text:

‘Patient and public involvement (PPI) is recommended from the earliest research stages through to dissemination of the study findings. (86) Patient and public involvement throughout the various stages of research is a valuable component of research activity and can contribute to improved quality and relevance of research. (87’)

Discussion

Please highlight the key findings in one sentence at the beginning of the discussion. Further details can be provided later. Given the complexity of the review, a point-by-point discussion is warranted, but first framed by a succinct topic sentence The Discussion section now includes a clear succinct sentence highlighting the key findings. This is followed by a point-by-point discussion of the key findings of the review. Key sentence at the beginning of the discussion section: 

This scoping review characterised the broad and varied unmet needs of patients living with a diagnosis of IPF. These unmet needs spanned five core domains incorporating physical and psychological needs, palliative care needs and finally needs related to information and health service provision (S5 File). 

Psychological distress was heightened...why does this represent a critical issue in your words. The Discussion section now clearly delineates why psychological distress is a critical issue: 

‘The psycho-social needs of patients and their family carers represent a critical issue as regards the incorporation of psychological support into existing and newly developed clinical care pathways for patients with IPF, with a focus on eliminating cost and access issues to this important resource. The literature supports a broad scope of adjunct psychological supports for patients including the requirement for a respiratory nurse specialist, peer support programmes and pulmonary rehabilitation. (38) However, there remains continued disparate access to these resources internationally. (38, 40)’

The findings of the review extend our current understanding...please explain further as to the specific met demands of the scoping review, e.g. characterizing X, identifying X, etc The Discussion section now clearly states: 

‘In summary, the findings of this review extend our current understanding of the broad scope of unmet needs for patients with IPF, expanding our depth of awareness of patient experiences and requirements for support and access to essential services across multiple domains. Equally it identifies the diverse and nuanced approach to care which is required for this patient population. Furthermore, it identifies an urgent unmet need globally which is the call for the establishment of IPF patient registries to capture epidemiological information, (44, 66), coupled with additional benefits such as ease of identification of patients who may wish to participate in research or clinical trials related to IPF. (18, 22) Registry data will offer opportunities to improve our understanding of this complex condition and to adequately prepare for appropriately resourced care provision’.

Reviewer #1 comments 

Eligibility criteria:

“The search was limited to human participants >18 years of age at the time of data collection (PCC Framework for search strategy development is available in appendix 1).”

IPF is disease of elderly, why the age >18 years set? Will it be better to replace it “adult”? Or studies that included young patient with pulmonary fibrosis were also included? This is important as the study title mention “IPF”. This observation is noted and the eligibility criteria has been updated to reflect your comments. Note that the study is related to patients with IPF, however given the similarities in symptoms and disease trajectory for some we also included studies that referred to patients with pulmonary fibrosis, but these studies had to also include reference to IPF. 

The age for being defined as an adult may differ from one international jurisdiction to the next, therefore the word adult and age were used together to overcome this potential issue.

A table has now been inserted to include inclusion and exclusion criteria (table 3 and table 4)

We utilised the ‘PCC’ framework, population (adult patients with IPF), concept (unmet needs), and context (all healthcare settings) to define the search strategy inclusion criteria. (25) The PCC framework used to inform the search strategy is presented in Table 2. The ‘PCC’ mnemonic (population, concept, and context) is recommended as a guide to construct clear objectives and eligibility criteria for scoping review. (26) The core concept in this review is ‘unmet needs’. Studies describing the unmet needs specifically of patients diagnosed with IPF were included. Inclusion and exclusion criteria are presented in Table 3 & 4. 

2.2 Scoping Review of the evidence

“This review was conducted in accordance with the Joanna Briggs Institute framework for scoping reviews and included the following steps: (i)identifying the research question, (ii)developing a search strategy, (iii)study selection and (iv)data analysis and presentation. (22)”

The most important aspect of the systematic review is the identification/framing of research question. Therefore, authors should provide the information related to research question. How the research questions identified/framed? Who were involved in designing in research questions? Were there any modifications in research questions? If yes, how many revisions? The section now clearly states: 

This review was guided by a central question, which was to map the available evidence related to the unmet needs of patients living with a diagnosis of IPF. The central research question was developed after several meetings with patient and public partners (PPI) comprising of patients diagnosed with IPF, their carers and healthcare professionals who collectively (a) described their research priorities and (b) identified the multi-dimensional component of their unmet needs. This was an iterative process and over the course of three meetings the research question took shape and led to the development of the scoping review protocol. A glossary is presented in S1 File.

2.3 Sources & searching

“Further, a comprehensive online search of key websites, a manual search of the reference lists of included studies and a search of annual conference abstracts sought to identify studies that may have been missed within the initial search.”

Were all annual conferences searched? Or conferences were specific to Pulmonary Diseases, ILD, IPF? Also, were these only regional, national or international conferences?

How many abstracts were screened/included/excluded?

 A clearer overview of the abstracts included from annual scientific conferences:

A representative snapshot of conferences delineated by the authors that regularly included research on interstitial lung disease and encompassing IPF research priorities were considered as part of the grey literature review. A total of 80 international conference abstracts were reviewed including those from the Irish Thoracic Society, the European Respiratory Society and the American Thoracic Society meetings. Nine abstracts were included in the final output.

‘Further, a comprehensive online search of key websites and a manual search of the reference lists of included studies was performed. Several international conference abstracts were reviewed including those from the Irish thoracic society annual scientific meeting, the European respiratory society annual meeting and the British thoracic society meetings. A total of 100 abstracts were reviewed with 9 abstracts included in the final review’. 

3.1 Summary of the studies

“All included studies were published in English and were from a wide geographical area (figure 2).”

There was no language barrier. However, authors did not report number of publications in language other than English. Note to the reviewer: The total number of non-English articles that the search revealed was five. Two of these were brought forward for full text review. Google translate was used to translate the artless and reviewed by a Spanish translator. When compared to the inclusion and exclusion criteria these articles were not included in the final 52 included citations. It should be noted that the author believes that a diverse range of sources were included in the review most notably the fact that the included citations hail from a wide geographic spread with many countries included where English is not the dominant language spoken. It was important to the authors to include a representative sample of the research being undertaken worldwide form both English and non- English language countries.

The intext reference to this is highlighted here:

After eliminating duplications, 884 unique citations were identified with five of these representing non-English sources. Of these, 719 records were excluded after title/abstract screening, leaving 165 records for further assessment. In addition, a further 20 records were identified through reference checks of systematic and narrative reviews. After full text review, 113 records were deemed ineligible and were excluded. The primary reason for exclusion was the lack of data on the outcome of interest (n=74). Other common reasons for exclusion were wrong population (n=11) and commentary or opinion piece (n=16). Fifty-two information sources met the inclusion criteria, of which (n=30/58%) were published in the last five years. All included articles were published between the period of 2011 to 2022. Included sources were published in English and were from a wide geographical area, including several countries where English is not the dominant spoken language and presented in Fig. 2. 

Page 6; para 2: “Several literature reviews (n=6) incorporated a range of methods. The remaining information sources were diverse (n=8) including guideline documents (n=2), patient charters (2), framework documents (n=1) reports (n=2) and a position statement (n=1). Information sources originated from multiple sources.” The text seems misplaced. The text had been updated with your comments: 

Several literature reviews were included (n=6) and incorporated a range of methods. The remaining information sources (n=8) were diverse and included guideline documents (n=2), patient charters (n=2), framework documents (n=1) reports (n=2) and a position statement (n=1). 

“Kalluri and colleagues in their qualitative study revealed that advance care planning is desired by patients and caregivers early in their illness experience with HCPs citing a need to clarify role, scope and responsibility with a call for practical guidance and training for HCPS to improve competency and confidence in these conversations.” Please provide the reference of the study. Kalluri and colleagues in their qualitative study revealed that advance care planning is desired by patients and caregivers early in their illness experience, with healthcare professionals citing a need to clarify role, scope and responsibility with a call for practical guidance and training for healthcare professionals to improve competency and confidence in these conversations. (53)

“This review included non-English language articles but cannot claim to have exhausted all non-English resources despite utilising a systematic approach to the review.”

However, there is no mention of the number of non-English studies in the manuscript. Also, if non-English studies were not excluded then how data extracted from those studies? Did authors take help from foreign language expert/s? If yes, please mention in the manuscript. Note to the reviewer: The total number of non-English articles that the search revealed was five. Two of these were brought forward for full text review. Google translate was used to translate the artless and reviewed by a Spanish translator. When compared to the inclusion and exclusion criteria these articles were not included in the final 52 citations included in the review. It should be noted that the author beloves that a diverse range of sources were included in the review most notably the fact that the included citations hail from a wide geographic spread with many countries included where English is not the dominant language spoken. It was important to the authors to include a representative sample of the research being undertaken worldwide form both English and English language countries and so non-English languages were included in the search strategy to facilitate this.

The intext reference to this is highlighted here:

After eliminating duplications, 884 unique citations were identified with five of these representing non-English sources. Of these, 719 records were excluded after title/abstract screening, leaving 165 records for further assessment. In addition, a further 20 records were identified through reference checks of systematic and narrative reviews. After full text review, 113 records were deemed ineligible and were excluded. The primary reason for exclusion was the lack of data on the outcome of interest (n=74). Other common reasons for exclusion were wrong population (n=11) and commentary or opinion piece (n=16). Fifty-two information sources met the inclusion criteria, of which (n=30/58%) were published in the last five years. All included articles were published between the period of 2011 to 2022. Included sources were published in English and were from a wide geographical area, including several countries where English is not the dominant spoken language and presented in Fig. 2. 

Reviewer Number 2: 

1- The inclusion of other reviews, it would be of more benefit to include studies from those reviews that are relevant to this current review Thank you for your comments. The JBI guidelines on scoping reviews state that scoping reviews are amenable to the inclusion of all methodologies…and may include evidence synthesise such as systematic reviews. Therefore, the author included other reviews in this scoping review in keeping with instruction from JBI scoping review guidelines.

2- The exclusion criteria are not specified so it is unclear how 1442 were filtered out. We have now included the following information: 

‘A total of 1492 information sources were identified through our database search, with 718 duplicates removed through manual and automated tools. After eliminating duplications, 884 unique citations were identified with five of these representing non-English sources. Of these, 719 records were excluded after title/abstract screening, leaving 165 records for further assessment. In addition, a further 20 records were identified through reference checks of systematic and narrative reviews’.

3- The search terms in appendix 2 would help to understand point 2 if they were included in the manuscript. Thank you, the manuscript is amended and now includes the following: 

An example of a search completed on CINAHL (EBSCO) of the search terms to identify the population is included here, using the ‘PCC’ acronym, and specifically looking at the ‘population’, (MH "Idiopathic Pulmonary Fibrosis") OR (MH "Idiopathic Interstitial Pneumonias+") OR (MH "Pulmonary Fibrosis+") OR TI ( “Idiopathic pulmonary fibros*” OR “Idiopathic interstitial pneumonia*” OR “Familial Idiopathic Pulmonary Fibrosis*” OR “Usual Interstitial Pneumon*” OR “fibrosing interstitial lung disease” OR “progressive fibrosis” OR “nonspecific interstitial pneumonia” OR “pulmonary fibros*” ) OR AB ( “Idiopathic pulmonary fibros*” OR “Idiopathic interstitial pneumonia*” OR “Familial Idiopathic Pulmonary Fibrosis*” OR “Usual Interstitial Pneumon*” OR “fibrosing interstitial lung disease” OR “progressive fibrosis” OR “nonspecific interstitial pneumonia” OR “pulmonary fibros*”). 

Reviewer no. 3 

1) Can the authors provide a clear outline, perhaps in tabular format, of the inclusion and exclusion criteria for publications to be included in the study, as this remains unclear.

 The manuscript now includes the following:

We utilised the ‘PCC’ framework, population (adult patients with IPF), concept (unmet needs), and context (all healthcare settings) to define the search strategy inclusion criteria. (25) The PCC framework used to inform the search strategy is presented in Table 2. The ‘PCC’ mnemonic (population, concept, and context) is recommended as a guide to construct clear objectives and eligibility criteria for scoping review. (26) The core concept in this review is ‘unmet needs’. Studies describing the unmet needs specifically of patients diagnosed with IPF were included. Inclusion and exclusion criteria are presented in Table 3 & 4. 

Table 2: PCC Framework for search strategy development

Framework Element Key Terms

PCC Population Adults > 18 years of age who have a diagnosis of idiopathic pulmonary fibrosis. 

 Concept Healthcare needs.

 Context All care settings.

Table 3: Scoping review inclusion criteria

Inclusion Criteria: Sources must relate to patients with IPF.

 Sources must be published between 2011-to 2022. 

 Adults >18 years of age.

 All geographic areas.

 All care settings.

 There are no language limits. Google translate will be used to translate non-English sources. 

 We will include all review types, including systematic, scoping and literature reviews, which describe the unmet needs of our patient group. 

Table 4: Scoping review exclusion criteria 

Exclusion criteria: Sources published prior to 2011.

 Refers to human subjects < 18 years of age.

 Does not include reference to patients diagnosed with IPF.

 This review will not include case reports, protocols, letters, posters, commentaries or opinion pieces.

2) The study cohort should be better defined. The manuscript refers to patients with IPF and PF in various sections, and in the results section also includes ILD (presumably non fibrotic)? The population being studied therefore is unclear and makes interpretation difficult. The population under investigation are patients with IPF. 

Table 2: PCC Framework for search strategy development

Framework Element Key Terms

PCC Population Adults > 18 years of age who have a diagnosis of idiopathic pulmonary fibrosis. 

 Concept Healthcare needs.

 Context All care settings.

3) Can the authors clarify why only English language publications were ultimately included in the study? Note to the reviewer: The total number of non-English articles that the search revealed was five. Two of these were brought forward for full text review. Google translate was used to translate the artless and reviewed by a Spanish translator. When compared to the inclusion and exclusion criteria these articles were not included in the final 52 citations included in the review. It should be noted that the author beloves that a diverse range of sources were included in the review most notably the fact that the included citations hail from a wide geographic spread with many countries included where English is not the dominant language spoken. It was important to the authors to include a representative sample of the research being undertaken worldwide form both English and English language countries and so non-English languages were included in the search strategy to facilitate this.

The intext reference to this is highlighted here:

All included articles were published between the period of 2011 to 2022. Included sources were published in English and were from a wide geographical area, including several countries where English is not the dominant spoken language and presented in Fig. 2. 

4) The description in the results section relates to patient and carer participation in study oversight seems better suited for inclusion in the methods. We have moved the PPI (patient and public inclusion) section to the methods section.

5) Finally, the manuscript should be reviewed for grammatical errors (particularly the use of commas), which seem to be present throughout and detract from an otherwise very strong and well-presented manuscript. Review of manuscript to correct grammatical errors has been completed.

---

## [Decision Letter · Decision Letter 1]

20 Dec 2023

PONE-D-23-27557R1A scoping review of the unmet needs of patients diagnosed with idiopathic pulmonary fibrosis (IPF).PLOS ONE

Dear Dr. Bramhill,

Thank you for submitting your manuscript to PLOS ONE. After careful consideration, we feel that it has merit but does not fully meet PLOS ONE’s publication criteria as it currently stands. Therefore, we invite you to submit a revised version of the manuscript that addresses the points raised during the review process.

Please submit your revised manuscript by Feb 03 2024 11:59PM. If you will need more time than this to complete your revisions, please reply to this message or contact the journal office at plosone@plos.org. Please include the following items when submitting your revised manuscript:A rebuttal letter that responds to each point raised by the academic editor and reviewer(s). You should upload this letter as a separate file labeled 'Response to Reviewers'.A marked-up copy of your manuscript that highlights changes made to the original version. You should upload this as a separate file labeled 'Revised Manuscript with Track Changes'.An unmarked version of your revised paper without tracked changes. You should upload this as a separate file labeled 'Manuscript'.If applicable, we recommend that you deposit your laboratory protocols in protocols.io to enhance the reproducibility of your results. Protocols.io assigns your protocol its own identifier (DOI) so that it can be cited independently in the future. For instructions see: https://journals.plos.org/plosone/s/submission-guidelines#loc-laboratory-protocols. Additionally, PLOS ONE offers an option for publishing peer-reviewed Lab Protocol articles, which describe protocols hosted on protocols.io. Read more information on sharing protocols at https://plos.org/protocols?utm_medium=editorial-email&utm_source=authorletters&utm_campaign=protocols.

We look forward to receiving your revised manuscript.

Kind regards,

Zyad James Carr, MD

Academic Editor

PLOS ONE

Journal Requirements:

Additional Editor Comments:

To the authors:

Reviewer has brought up an important point regarding certain biases related to abstract selection. Please address these comments and return for final approval. This will likely require that the co-authors clearly state w/in the methodology that North American abstracts were not reviewed, that the scoping review was limited to European sources, and should be noted consistently in the title and body.  The other alternative is expand your review to include abstracts from North/South America and Asia. Please be clear and concise regarding the methodology and pool of applicable data that was utilized for the scoping review. 

Reviewer 2:

There is a contradiction between the exclusion criteria for posters (Table 4), which are presented as abstracts at scientific meetings, and the inclusion of abstracts as stated on page 9: "Several international meetings were reviewed, including the Irish Thoracic Society Annual Scientific Meeting. abstracts were reviewed, including those from the Irish Thoracic Society Annual Scientific Meeting, the European Respiratory Society Annual Meeting, and the British Thoracic Society meetings. A total of 100 abstracts were reviewed and 9 abstracts were selected for final review."

Thank you

Reviewers' comments:

Reviewer's Responses to Questions

**Comments to the Author**

1. If the authors have adequately addressed your comments raised in a previous round of review and you feel that this manuscript is now acceptable for publication, you may indicate that here to bypass the “Comments to the Author” section, enter your conflict of interest statement in the “Confidential to Editor” section, and submit your "Accept" recommendation.

Reviewer #2: All comments have been addressed

Reviewer #3: All comments have been addressed

2. Is the manuscript technically sound, and do the data support the conclusions?

Reviewer #2: Partly

Reviewer #3: Yes

3. Has the statistical analysis been performed appropriately and rigorously? 

Reviewer #2: (No Response)

Reviewer #3: Yes

4. Have the authors made all data underlying the findings in their manuscript fully available?

Reviewer #2: Yes

Reviewer #3: Yes

5. Is the manuscript presented in an intelligible fashion and written in standard English?

Reviewer #2: Yes

Reviewer #3: Yes

6. Review Comments to the Author

Reviewer #2: There is a contradiction between the exclusion criteria for posters (Table 4), which are presented as abstracts at scientific meetings, and the inclusion of abstracts as stated on page 9: "Several international meetings were reviewed, including the Irish Thoracic Society Annual Scientific Meeting. abstracts were reviewed, including those from the Irish Thoracic Society Annual Scientific Meeting, the European Respiratory Society Annual Meeting, and the British Thoracic Society meetings. A total of 100 abstracts were reviewed and 9 abstracts were selected for final review."

Reviewer #3: The authors have addressed all questions from the editors and reviewers - I am not sure why there is a minimum word count for this question

7. PLOS authors have the option to publish the peer review history of their article (what does this mean?). If published, this will include your full peer review and any attached files.

Reviewer #2: No

Reviewer #3: No

---

## [Author Response · Author response to Decision Letter 1]

4 Jan 2024

Dear Editorial team and reviewers,

I want to sincerely thank you for your time in reviewing our scoping review. Please find out comments below. 

Reviewer comment: Reviewer has brought up an important point regarding certain biases related to abstract selection. Please address these comments and return for final approval. This will likely require that the co-authors clearly state w/in the methodology that North American abstracts were not reviewed, that the scoping review was limited to European sources, and should be noted consistently in the title and body. The other alternative is expand your review to include abstracts from North/South America and Asia. Please be clear and concise regarding the methodology and pool of applicable data that was utilized for the scoping review

Reply: Thank you for your comment. This list of conference abstracts was not intended to be exhaustive but rather a reflection of conferences abstracts that were reviewed as part of the systematic search of the literature as part of this extensive scoping review.

There were no geographic limits placed on the search strategy in this extensive review. 

In the first instance in the initial search of databases included in the search strategy abstracts were returned along with full text sources and the search was not limited to any particular geographic area. 

In the second instance in the conference abstract search the American thoracic society abstract booklet was also reviewed as part of the scoping review.

We did not search conferences in Asia or South America specifically, but we are satisfied that based on the initial search which did not have geographic limits that we will have facilitated scope to include available literature including abstracts from those areas. 

Reviewer comment: There is a contradiction between the exclusion criteria for posters (Table 4), which are presented as abstracts at scientific meetings, and the inclusion of abstracts as stated on page 9: "Several international meetings were reviewed, including the Irish Thoracic Society Annual Scientific Meeting. abstracts were reviewed, including those from the Irish Thoracic Society Annual Scientific Meeting, the European Respiratory Society Annual Meeting, and the British Thoracic Society meetings. A total of 100 abstracts were reviewed and 9 abstracts were selected for final review." 

Reply: Thank you for your comments: 

Table 4 now includes the following text: ‘This review will not include case reports, protocols, letters, commentaries or opinion pieces’.

Page 9 now included the following text: ‘Several international conference abstracts were reviewed including those from the Irish thoracic society annual scientific meeting, the European respiratory society annual meeting, the British thoracic society meetings and the American thoracic society meeting. A total of 100 abstracts were reviewed with 9 abstracts included in the final review’.

Many thanks,

Carita

---

## [Editor Report · Decision Letter 2]

15 Jan 2024

A scoping review of the unmet needs of patients diagnosed with idiopathic pulmonary fibrosis (IPF).

PONE-D-23-27557R2

Dear Dr. Bramhill,

We’re pleased to inform you that your manuscript has been judged scientifically suitable for publication and will be formally accepted for publication once it meets all outstanding technical requirements.

Kind regards,

Zyad James Carr, MD

Academic Editor

PLOS ONE

Additional Editor Comments (optional):

Although methodological flaws are still present, consideration of the breadth and depth of the chosen topic should be accommodated. Thus, I hope you continue your scientific work to include original contributions on the identified gaps that you and your co-authors have found in your scoping review. 

Best Wishes,

Zyad J. Carr, M.D., FASA

Associate Professor of Anesthesiology

Division of Critical Care Medicine

Yale University, School of Medicine
---

## [Editor Report · Acceptance letter]

31 Jan 2024

PONE-D-23-27557R2 

PLOS ONE

Dear Dr. Bramhill, 

I'm pleased to inform you that your manuscript has been deemed suitable for publication in PLOS ONE. Congratulations! Your manuscript is now being handed over to our production team.

Kind regards, 

on behalf of

Dr. Zyad James Carr 

Academic Editor

PLOS ONE